# Privacy, Property, and Third-Party Esteem in Arendt's Constitutionalism

**Emmett McGroarty** [1,*] **and Brendan McGroarty** [2]

1    Honors College, Belmont Abbey College, Belmont, NC 28012, USA
2    Independent Researcher, Portage, MI 49002, USA; brendan.mcgroarty@gmail.com
*    Correspondence: emmettmcgroarty@bac.edu

**Abstract:** In *On Revolution*, Hannah Arendt makes the case that a constitution must account for the need of the human person to participate in the building of society, both as a primordial and continual action of founding. This paper draws on Arendt's insight on the relationship between privacy and the notion of property, both of which the constitution must protect, as it is dependent on those notions. Property in its fullest sense is the means by which a person interacts with others and establishes a society. Particularly important for this notion of engagement are the concepts of shame and the love of goodness. The actor emerges from the private sphere to interact with others on the strength of the secrecy and confidentiality of her intimate, private relationships. Property is therefore essential to human flourishing and happiness. Following this, the activity of constructing the public forum on the basis of the private is an important feature of Arendt's constitutionalism. Human Action showers third-party esteem on the actor's family and friends, binding them to the constitutional structure and strengthening familial relationships and social cohesion.

**Keywords:** Arendt; constitutionalism; property; privacy; esteem; human condition; public freedom; Human Action; Tocqueville; totalitarianism; tyranny; third-party esteem; shame

## 1. Introduction: Arendt's Constitutionalism Fosters the Pursuit of Happiness

In the epilogue to his memoir *A Confession*, Leo Tolstoy describes a dream in which he suddenly becomes aware of resting, suspended between an unfathomable abyss below and an infinite space above (Tolstoy 2009, pp. 2196–99). As he looks downward into the abyss, he feels terror and anxiety, a slipping all the more acute as he struggles to arrest his fall. Yet, when he looks to the infinite space above, a soothing calm settles in him. He recognizes from an Archimedean point that he is secure and that his orientation to the infinity above properly reflects his nature. Tolstoy recognizes himself as fundamentally making one of two choices: turn inward toward the safety of certain death and limitation, or outward, beyond the limitations of his society toward the source of infinite creativity.

Hannah Arendt avoids making systematic claims about human nature, shunning even the title *philosopher* as evocative of the twentieth century contemplatives who dreamed while tyrants readied the world for living hell (Arendt 2018, pp. 9–11). She instead describes the activity of being human that distinguishes us from other animals as humans gaze toward the infinite, relating with others to achieve greatness beyond themselves. Humans engage in the contemplative life and in the active life, the latter with varying degrees of freedom which she describes as labor, work, or action (Arendt 2018, pp. 7–8, 14–16). Labor consists of those activities that must be undertaken to sustain life. Work advances beyond the necessary to works of artifice. Human Action encompasses those activities necessarily completed through encounters with other persons; "only action is entirely dependent upon the constant presence of others" (Arendt 2018, p. 22). Only through action, that is, interacting with others to create something new and lasting beyond themselves, can humans find fulfillment or happiness. For Arendt, constitutionalism was not merely the

control of government, but the creation of power and authority of the people themselves which was necessary for human happiness. Fulfillment, or happiness, is a central concern of constitutionalism, and this occurs only through active participation.

Whether through the active or contemplative life, humans seek happiness. Arendt sets aside the contemplative ideal as well as a definitive reflection on human nature itself. Instead, she studies the orientation the human person has toward the other. Happiness lies first in liberty from necessity and ultimately in freedom to act, that is interact with other persons. Humans satisfy the conditions of survival in order to be free to interact publicly. While in this paper we do not offer any definition of human nature, we do posit two innate characteristics consonant with Arendt's thought: humans are drawn to interact with one another in the process of knowing the self and the other, and humans seek to shape their private and public environment. A person seeks fulfillment by interacting with other persons, acting in concert with the other, and defining their world through the creativity of speech (Arendt 2018, pp. 25–26). All humans do this in their various and unique ways.

For Arendt, happiness lies in this orientation toward a greater life than can be realized in an individual life. Private life, rather than being a diminution of happiness, forms the bedrock of public life. What is stored and cultivated in private is protected from the public for the sake of cultivation and self-reflection. What is private may then be shared or drawn upon when persons freely create a public space, independent of themselves and their interests. They create a forum of surpassing beauty through common consent and the inherent power of speech (Arendt 2006, pp. 9, 109–11). While the private sphere is foundational and takes logical precedence, human nature finds its teleological fulfillment properly in the public sphere. We contend that property transcends these two spheres, understanding that property comprises everything proper to the person, including ideas, beliefs, identities, relationships, speech, and physical material. Physical property, like the physical environment, often serves as a metaphor for all other types of property but should not be considered a limit.[1] Similarly, references to environment allude not only to the physical environment, but the context, public or private, in which property is held, used, and disposed; culture, history, language, landscape, and intellectual and legal heritage all form part of the environment of property. As such, property is the subject of constitutionalism.

In this paper, we address an aspect of the growing problem of civil discord and alienation from government.[2] We suggest that Arendt's insight into the practice of being human provides civil society, and specifically constitutional practice, with important, yet underappreciated, insights on tranquility and happiness. In a nutshell, a constitutional structure should provide a platform for Human Action; that platform should extend a universal invitation to the citizenry, and it should cleave successive generations to the constitutional project. We contend that the failure to recognize these purposes has severe social consequences. We also contend that the tendency to overlook it produces an impoverished understanding of constitutionalism.

The next section will explain the importance of property, a concept which has been used in the past as a means of disqualifying people from participating in government. However, we seek to rehabilitate this term as a useful concept to account for the importance of privacy and the dignity of the person. Since the recognition of intellectual property and the concept of data as property, the notion has come to mean everything proper to or belonging to a person, from which that person may choose to relate to other persons.

---

1   Hegel has a similarly expansive view of property. See (Ritter 2004, p. 112).

2   From 1958 through the early 1960s, the percentage of citizens with a high level of trust in government hovered in the 70s. It has declined steadily since then and for the last decade has sat in the high teens, according to data compiled by the Pew Research Center. Public Trust in Government: 1958–2021, https://www.pewresearch.org/politics/2021/05/17/public-trust-in-government-1958-2021/, accessed on 14 August 2023. Likewise, American citizens have grave concerns about the level of trust Americans have in each other. Lee Rainie, Scott Keeter, and Andrew Perrin, "Trust and Distrust in America", https://www.pewresearch.org/politics/2019/07/22/trust-and-distrust-in-america/, accessed on 14 August 2023.

This includes notions about how to govern and who should govern. However, the self is distinct from property; one does not own oneself and cannot disclose the self, only ideas of the self. The self remains shrouded in privacy along with whatever property is not made available to others or the public. The third section addresses Arendt's notion of the need all humans have to participate in public action. While humans first address necessities, their full happiness can only come from relating and interacting with others, participating in what she calls Human Action. The reach for the eternal is what distinguishes humans as such. Participating with others in public action can only come about through addressing the innate experience of shame in forming confidence in the self.

Alexis de Tocqueville's concept of association, in particular permanent association, is addressed in the fourth section. Permanent association is the locus in which the person most fully participates in public action and, in doing so, implicitly accepts the equality of all other members of that association. The fifth section addresses some of the problems which arise when the person is alienated from decision making, whether through tyranny or the unresponsiveness of elite administrators or politicians. The sixth section introduces the concept of third-party esteem, which refers to the process whereby networks of relationships encourage the full participation of a wide variety of people, and the benefits this bring to the person and to society. The experience of third-party esteem is essential for generating public trust, even where such trust is not justified by personal relationship. The seventh and final section before the conclusion describes how the centralization of government can sap the energy of its citizens, even when the central government simply seeks to aid and assist the local government by giving grants and setting stipulations.

## 2. Private Property Is Integral to the Human Person

The basis of Arendt's constitutionalism is the human person and the use of property to shape the environment and form interpersonal relationships. A constitutional architecture, if it is to be good, must be compatible with, if not provide conditions for, the individual's pursuit of happiness. In the active life, happiness arises from the outward interaction with others in the building of things that cannot be achieved by the individual alone. In a limited sense, the human person needs to associate with others to build physical projects that require labor. However, even for those projects, and certainly for other projects, the individual recognizes that "plurality is the condition of Human Action because we are all the same, that is, human, in such a way that nobody is ever the same as anyone else who ever lived, lives, or will live" (Arendt 2018, p. 8). If the human person is to build, she must recognize her own limitations and look toward others for their gifts. A constitutional scheme must take into account these dynamics.

Arendt grasps the importance of property to the human person, most especially to Human Action. Historically, this importance can be most starkly recognized in the consideration of real property (land and the improvements made on it). Within the clear-cut boundaries of real property, the fee holder (e.g., owner or lessee) can bar the intrusion of others, whether those be government or private parties. There, she sets the rules, defining protocol and granting admission.[3]

Arendt's notion of property extends beyond that which may be commoditized as wealth. Hegel also believes property extends beyond the mere physical, encompassing our faculties, ideas, relationships, etc., none of which can be completely disposed by the person (Ritter 2004, p. 112). However, the difference between Arendt's conception of property and that of Hegel is that Arendt's is much more profound, personalized, and, in some ways, concrete. For Arendt, property is always grounded in privacy. She describes how in the ancient world, property held a sacred character which could not be alienated from a person despite the vicissitudes of wealth because it was private, impenetrable to the public eye because one's property was the realm of life and death, and the public eye cannot

---

3   Historically, through real property the family maintained its connection with ancestors and conducted ceremonies of marriage, giving property a sacred character (Siedentop 2014).

comprehend the mysteries of life and death (Arendt 2018, pp. 60–62). From the perspective of the public, the four walls of property represent a legal barrier to privacy. However, from the perspective of the householder, property, as a manifestation of this hidden reality, is a good in itself.

For Hegel, while property is protected against fellow citizens, the state can have no such restriction (Wilson 2019, pp. 86–88). Property holds no sacred value. It is important because it is an instantiation of abstract right to property, and so makes the freedom which is inherent in the person manifest. Property may be taken from a person, but the person nonetheless holds abstract right to property. For Arendt, however, property is not abstract, even if not material. It is through particular property that the person interacts with others in public. My right to loiter in my neighborhood comes from the fact that I own property in my particular neighborhood, not another. My right to speak comes from my own self-conception and particular facility with language. For Hegel, the particular examples of property are incidental and a means to freedom; the freedom itself exists in abstract. For Arendt, the freedom one has is in the sacred property that person holds through which, in a concrete way, the person interacts with others.

Arendt's writings rest on a broad concept of property. In that regard, western legal tradition recognizes several forms of property. These include, for example, real property, personal property, and intellectual property. We submit that Arendt's constitutionalism calls for a broader concept of property than what is legally enforceable. Specifically, property extends to all things proper to oneself. Property includes all creations of the person. It includes relationships, which in Arendt's thought exist in the space between persons, and ideas. Again, not all forms of property are enforceable under the law, but all are in some sense a product of the person's creation, even if through purchase, gift, or abstract thought.

From her property, the human person gives to, and forms relationships with, others, which themselves give her a form of property. Human nature finds fulfillment in the establishment of these relationships, human and divine, and the formation of associations on a larger scale. Through those relationships the person shapes her private environment and exercises her share in shaping the public environment. That fulfillment is happiness and is the natural end of being human.

A person's property can be thought of as a personal reservoir from which one draws in a variety of ways. The pauper may have no real or personal property, and may not have, as Arendt would describe it, a location in the world, but nonetheless enters the public sphere with thoughts and relationships from which to act, even if such action is a struggle. Property helps the person understand herself and appreciate others. An appreciation of property inculcates in the person a sense of humility, or limitation, and an appreciation of the gifts of others. For all these reasons, property is highly personal and sacred.

The usurpation of property is an impairment on that person's exercised (past) liberty because it denies the person the fruits of that exercised liberty, and it is a taking of that person's future liberty because she can no longer use or dispose of the property. Furthermore, a narrow concept of property invites invasion of privacy. Frederick Douglass, writing to Thomas Auld, who had held him in slavery and at the time still held his siblings, explains that he took nothing of what belonged to Auld, but took with him only his own faculties. As he describes, almost apologetically, it was only because he retained privacy against Auld's efforts that escape was possible.[4]

---

4   This formulation of rights recognizes a seamless continuum from life through property, accentuating that all rights are embedded in the person. It takes into account, we contend, how the person naturally views himself. See, e.g., (Douglass 1848), ("You are a man, and so am I. God created both, and made us separate beings... In leaving you, I took nothing but what belonged to me, and in no way lessened your means for obtaining an honest living. Your faculties remained yours, and mine became useful to their rightful owner".); (Madison 2002) (Man has property not only in his "land, or merchandize, or money" but also in "his opinions and the free communication of them... He has a property of peculiar value in his religious opinions, and in the profession and practice dictated by them. He has a property very dear to him in the safety and liberty of his person".); (de Tocqueville 2002, vol. II, part IV, chp. 6) ("Above [the citizenry] arises an immense and tutelary power that alone takes charge of assuring their enjoyment and of looking after their fate.... This is how it makes the use of free will less useful and rarer every day; how it encloses the action of the will within a smaller

An expanded sense of property, concerning rights not emanating from material goods, allows the nature of law to remain consistent with previous tradition as well as legal developments such as intellectual property. Before the twentieth century, laws protected property, not freedom directly (Arendt 2006, p. 172). Once the rights of individuals who owned no physical property became recognized, laws had to protect these rights.

Property is a condition of the active life in both a negative sense and a positive sense. In the positive sense, property is that which guarantees freedom because it is the means of freedom, or interaction. In the negative sense, its boundary lines give notice to the outside world as to the realm where privacy must be respected. In this way, privacy is authority over property, which the person can attenuate and make public in varying degrees: As Arendt noted, "The four walls of one's private property offer the only reliable hiding place from the common public world" (Arendt 2018, p. 71).

Although most people would shudder to think of a world without recognition of privacy, the appreciation of privacy seems to be diminishing. Roger Berkowitz posits three reasons for this. One, with the advance of technology and the ensuing benefits, privacy is inconvenient. Two, privacy is viewed as dangerous, as shielding the criminal and the terrorist. Third, privacy reeks of being anti-democratic, as offering a means to segregate oneself from majority culture and opinion (Berkowitz 2016).

In the face of that disintegration, Arendt fleshes out the strong connection between property and privacy. In so doing, she lifts the concept of property up from denigration as an ignoble right that fosters greed and describes it as both a condition and product of human flourishing. Berkowitz points out that Hannah Arendt is "the last and perhaps only great political thinker who set privacy at the center of her thinking". Not surprisingly, across disciplines, the academic justifications of privacy are "confused and contradictory".

Drawing on Arendt, Berkowitz organizes the justifications for privacy into three categories. First, privacy secures the depth of who we are, our "exclusiveness and uniqueness as a person". It is the harbor for our private selves, the place from which we "grow and thrive from out of feelings, urges, lusts, rages, and whims that are unfit for public consumption". It is here that the selves that we are ashamed of sit in harbor, neither circulating in public nor subject to its invasion (Berkowitz 2016).

Second, privacy protects the ability to shape our lives. "Fulfilling our needs and wants are part of human flourishing". These needs and their fulfillment flow from initiative, and initiative is rooted in our private urges and desires. If the social and the public invade our private life, those drives and initiatives get truncated or obliterated as we respond to the social and the public.[5]

Third, privacy enables the judgment of taste. Through the activity of taste we decide how to present ourselves to the public, what others will see and hear, and what we want to remain private. In this way, privacy "protects the meaningfulness and substance of the public world".[6]

Taken together, the justifications for privacy enable the person to maintain her independence of thought and judgment, and property provides the realm in which privacy operates. In the private realm, the person can define her self, delineating the contributions she might make through Human Action.[7] There she can better evaluate her gifts or abilities and, by contrast, better recognize those of others; that dynamic draws the person to a sense of humility on one hand and affirmation of human dignity on the other.

In contrast, a lack of privacy enervates life and crushes the human spirit. When the social and the public invade the private, they deprive us of the chance to freely shape our

---

space and little by little steals from each citizen even the use of himself".). In both government and private action, the modern trend is to separate the right to property from the person and to thereby minimize it.

[5] Berkowitz.

[6] See Footer 5 above.

[7] (Arendt 2018, pp. 28–30): "What prevented the polis from violating the private lives of its citizens and made it hold sacred the boundaries surrounding each property was not respect for private property as we understand it, but the fact that without owning a house a man could not participate in the affairs of the world because he had no location in it which was properly his own".

lives and the lives of our children. As Berkowitz argues, "If we do not defend the rights of parents to raise their children in their personal and unique belief and value systems, what then does privacy mean?" Privacy enables the depth of personal difference, and its obliteration leads to conformity.[8]

But is there not a downside? "Meaningful privacy", as Berkowitz observes, "is always in contest with social and normal standards of mass society, standards of progress and conformity". In that vein, is not the private where dark thoughts are held, reprehensible views that undergirded practices such as segregation and slavery? The abbreviated answer is that sometimes those views are formed in private, and sometimes they are the result of invasion by the public and social, badgering the person to join the tyranny of the majority, for example.

The full answer, though, considers how the private interacts with the public. The goal of the tyrant (whether of the majority, the minority, the few, or the single person) is to dictate conformity. As Tocqueville quipped, he does not care so much whether you love him, he just wants that you will not love each other, and this qualitatively and quantitatively limits Human Action. The public and social are always tempted to invade the private to eliminate what they judge, rightly or wrongly, to be anti-social opinions and speech, but such invasions smack of coercion and arouse alienation from the external.[9]

The dynamics of the private counter the tyrant and militate toward recognition of others' dignity. There, we prepare for Human Action, our encounters with others. In those encounters, we learn about others and appreciate their points of view, even if we do not always agree with them. We learn to accept people for their essential differences, or gifts, and appreciate their accidental differences. Through the processes of shame, esteem, contempt, and third-party esteem, we judge what is good and plan how we will present ourselves to others. It is in these processes that we evaluate external norms and passions. That is our chance to struggle with those external factors, to distinguish the tyrannical from the good. In these ways, the private life protects the public while also testing itself.

Privacy and property cannot exist without one another, but they are not identical. Privacy is the condition for self-reflection and creativity in which property may be held or withheld from the public sphere. The ability to hold property in custody, no matter how passively acquired, is an act of creativity. The ability to withhold property and direct its use represents intentionality. Through privacy, the person can determine the way in which she chooses to shape not only her own environment, but the public environment. In that way, privacy is the condition of both property and the public sphere.

Privacy, though, extends deeper than just the environment for property. It is also the condition of solitude or the encounter of the self with the self (Arendt 1968, p. 476). The self is not property; it makes no sense to speak of the self as belonging to the self, and the self does not create the self. If the self were property, then one could sell oneself into slavery. The person does indeed define herself in determining how to appear in public, but this image of the self does not exhaust the undefinable self encountered in solitude. This image of self is cultivated and offered for exposure to the public or private disclosure to another within the realm of privacy, creating a vulnerability, because although this image itself is the property of the self, it requires recognition by others in order for public action to be possible. The image is offered, but the recognition is not of the image, but of the self hidden in privacy. The self retains the right to disclose the self in an image or mediate the self through its property, undamaged in dignity but always risking isolation by the refusal of the other to acknowledge her property and thereby the personhood of her self.

In the absence of privacy, in a life lived entirely in the public eye, a person could never attain what Arendt describes as "goodness" or any depth of meaning. Such a life must remain shallow and with no moral direction or value (Arendt 2018, p. 71). On the one hand, a person without property has no leisure to participate in public affairs, but on the other, a

---

8　　See Footer 5 above.
9　　See Footer 5 above.

person without property in an expanded sense of the term has no ideas or even sense of self to contribute; property is not possible without privacy. Arendt's concept of privacy accounts for the productive use of shame, which in this sense is a self-reflection that helps us to recognize how we appear in the eyes of others. Shame is necessarily private, though it is the beginning of the person's public consciousness and awareness of the divide between public and private. The totalitarian will seek to destroy Arendt's "four walls of privacy" and weaponize shame in order to stifle public participation. Shame is weaponized when it is deprived of privacy. Without the development of shame in privacy, the sense of the self is curtailed and the individual becomes a creature of the regime.

Property is an important means by which one relates to the world and forms essential relationships and associations. Property in a positive sense, like law in a negative sense, both guarantees and symbolizes the freedom of the person in these relationships (Arendt 2006, pp. 171–72). Without property and its integral privacy, the person would lose confidence in her thoughts, and they would become alien to her. A radical loss of property leaves the person with no privacy and a stunted sense of the self, making Human Action fragile. Such a radical loss can often be observed in the homeless and in the institutionalized.

Arendt's reflections on her life under Nazi domination and her eighteen years as a stateless refugee help flesh out the connection between privacy and property. The Nazis reversed the conditions of freedom for Human Action. As Arendt observed, "If there was a relationship between these two spheres [the household and the *polis*], it was a matter of course that the mastering of the necessities of life in the household was the condition for freedom of the *polis*" (Arendt 2018, pp. 30–31). As the Nazis intuitively realized, property must be stripped away from members of the *polis* in order to isolate them and eventually achieve the condition of individual loneliness in which totalitarianism may take root. While the Nazis may have allowed physical property, its use was so circumscribed that it hardly qualified as a means of mediating the self or contributing to public affairs. By stripping away privacy, property became impotent. The totalitarian seeks to deny the individual the platform on which she forms her own thoughts. As Arendt observed, "the iron band of total terror leaves no space for such private life and...the self-coercion of totalitarian logic destroys man's capacity for experience and thought just as certainly as his capacity for action" (Arendt 1968, p. 474).

Totalitarianism "is not content with this isolation and destroys private life as well" (Arendt 1968, p. 475). It is not enough merely to render the person unable to act because no one will act with her. Such a person could still have a sense of solitude, having the capacity to maintain an inner dialogue with the self and act "together with" herself in an independent manner (Arendt 1968, p. 476). The totalitarian recognizes that such solitude is a prerequisite for action, and that person is still a threat. The totalitarian state thus requires loneliness, or the feeling of being deserted by all human companionship, even the companionship of oneself (Arendt 1968, pp. 474–76). To put it another way, the totalitarian seeks to deny the plurality of the human condition and to thereby dictate conformity.

For Arendt, goodness necessitates pure intentions, and the intrinsic goodness of a person is necessarily shrouded in privacy (Arendt 2018, p. 74). The seat of self-reflection and consciousness is in the private realm, and there the root of public action lies. "Goodness can exist only when it is not perceived, not even by its author"; once an action is public, then it is difficult to evaluate the quality of the actor (Arendt 2018, p. 74). Once a person becomes conscious of her goodness then she may doubt her own motivation. She becomes aware of her actions as others may see them, and so the original luster of goodness is lost since the motivation can no longer confidently be called good. Likewise, once a good action becomes public, it no longer makes sense to call it good. It can be recognized as practical, useful, or charitable, but to call it good means to ignore the self-consciousness of the author. Additionally, public actions are no longer personal or private, which is the only perspective that can perceive moral goodness. However, even though for Arendt goodness must remain private, public action is contingent upon the moral integrity of the other which is implied in the concept of recognition (Iser 2019). If goodness cannot

be assumed, all public action can only proceed from self-interest and commoditization of relationships, rendering any constitution impotent.

In democratic forms of government, which are predicated on freedom, one person must assume the goodness of another in order to act in concert with her. In other words, a prerequisite for Human Action is trust. For persons to join together in Human Action, they must trust one another, the minimum depth of which varies depending on the underlying activity. The public forum makes no assumptions about the intrinsic goodness of persons; rather, particular societies and their successors implicitly agree to trust the intentions of each for the mutual benefit of all. In that way, the American colonists flourished in the formation of power through "the then newly discovered means of promise and covenant", which, we note, led eventually to the Constitution (Arendt 2006, p. 167). In the public domain, goodness must be a general belief of others in the forum, a condition that is strengthened through the dynamic of third-party esteem, discussed below. When this general belief is lacking, Human Action is in peril. Thus, according to Arendt, the greatest threat to human existence "is not the abolition of private ownership of wealth but the abolition of private property in the sense of a tangible, worldly place of one's own". (Arendt 2018, p. 70). Wealth is the commoditization of property, or the reduction of property from how it is appreciated in private, and the meaning it might have, to a colorless medium that can be usurped and exchanged for the benefit of another. Property is a threat to the tyrant, ownership of wealth is not. Property requires trust in exchange; wealth does not. Property has profound importance, one much more personal than mere wealth. In contrast, tyrants do not require trust, because action is compelled. Trust implies consent to the expression of power because it also implies participation in that power.

Arendt observed that contemporary society was shifting away from real estate and favoring more fleeting forms of property. This change dulls individuals to the distinctions between the private and public sphere. Paradoxically, as Western civilization seems to celebrate the individual, it hollows out the goodness of the person and evermore devises political, contractual, and technological ways to invade the private space of the individual. People are losing their appreciation of privacy and recklessly making their property open to others, if not the public at large. Superficial interaction with others replaces creative action. The public demands thought conformity, bit-by-bit stealing from the person even the use of herself. This is a problem because at root a constitution provides for the interaction of persons for the public good. A materialist might reduce these interactions to the contribution of wealth to the treasury, and this wealth requires little or no privacy. However, more significantly than wealth, the poor man can contribute by participating in that action, creatively lending his unique perspective and voice, which can only emerge if a genuine sense of privacy of the person is recognized.

Arendt's insight into property as critical to Human Action demands a systematic approach to individual rights, one that encompasses the notion of property as integral to the human person. Property is now popularly viewed as a secondary right, as struggling to stay within the canon of fundamental rights. Perhaps this is partly due to it being a common target for taxation, whereby it is reduced to monetary quantification. Perhaps it is partly due to society itself being much more mobile than in centuries past, lending to a diminution in the significance of both real and personal property. As Arendt makes clear, property cannot be divorced from the human person, but must be viewed as integral to human activity. We propose a continuum for considering human rights that recognizes the integration of property rights in the human person. A person has the right to life, which inheres in one's existence but must be maintained through property. One acts, or exercises liberty, in the pursuit of happiness which involves interaction with others, whether publicly or privately. A goal of a constitution is the protection of property as a medium for the exercise of freedom, which for Arendt is public action, and as a reservoir of liberty exercised.

### 3. The Object of Human Action Is Public Happiness

Building on British tradition, American colonists further developed the concept of happiness. Its pursuit is the goal of life and includes both public and private aspects. Private pursuits mean the shaping of one's immediate environment: one's decisions about life, family, work, etc. The pursuit of public happiness means the exercise of political power in the public square which in a democracy[10] must be enacted in conjunction with others and in recognition of their political freedom. Arendt contends the American revolutionaries were not fighting for material justice or private happiness, but for public happiness, and this conception of happiness animated the constitutional process. Nonetheless, privacy must protect property in order to ensure the possibility of public action, and public action has a similar obligation to protect privacy (Arendt 2006, p. 121).

Acknowledgment of freedom encompasses the right to pursue both individual and common ends. As Hannah Arendt posited: "Americans knew that public freedom consisted of having a share in public business, and that the activities connected with this business by no means constituted a burden but gave those who discharged them in public a feeling of happiness they could acquire nowhere else" (Arendt 2006, p. 110). Happiness in this fullest sense is not derived as an effect of power over another person. Nor is it a consequence of material satiety. Happiness is its own goal and is the product of participating in action for the good of others and oneself. In Arendt's framework, informed by her reflections on the American Experiment among other things, public happiness is pursued through Human Action. A constitution, then, though it must circumscribe tyranny, ought to be oriented to the promotion of Human Action. Though citizens may in fact be content with private happiness, the republican constitutional system is oriented toward the full participation of its citizens, because full participation generates a plenitude of power; just as citizens can acquire happiness that comes from public freedom nowhere else, the power and authority of a fully engaged public exercising freedom can come about in no other way.

The colonies and the new republic, as well as any republic ever formed, faced obvious difficulties with its claim or the assumption of equality among the citizens. Perhaps in a theocracy, where all citizens are equally subject to a transcendent and arbitrary ruler, this could make sense. Yet, how is it claimed that citizens of vastly different wealth, background, inclination, education, and training are equal? What might equality mean in this contest? This question is particularly pertinent in a society that insists on the inviolability of privacy. Typically, the question is answered as an equality of rights that inhere in the person, which historically rely on natural law claims about the sanctity of the human person. While this claim holds validity, we propose that the fundamental right of privacy, which protects property, guarantees the ability to exercise public freedom and is a source of equality in a practical sense.

Natural law, history, and psychology affirm the need to pursue public happiness. The opportunity for free *participation flows from political equality*. As Tocqueville noted, where such equality exists, the person must reach out to others to garner their support (de Tocqueville 2002, pp. 485–88 [vol. 2, part 2, chp. 4]). In doing this, one is drawn to treat that person as an equal and to try to understand her point of view. That dynamic quashes the classification of others as unequal based on accidental differences such as race, geography, sex, religious practice, wealth, etc. It also draws one away from the collectivist, elitist, and conflict theories that have plagued modernity.

As a starting point, we assume the human person is free, rational, sociable and has a sense of her moral equality with others (Locke 1990, pp. 269–72, 308–9 [§§4–7, 61–63]; Barnes 1923). She possesses speech, which serves to:

> set forth the expedient and inexpedient, and therefore likewise the just and the unjust. And it is a characteristic of man that he alone has any sense of good and

---

10 We use the term *democracy* broadly and intend for it to encompass the constitutional republic instituted through the Constitution.

> evil, of just and unjust, and the like, and the association of living beings who have
> this sense makes a family and a state. (Aristotle 1885)

The human person yearns to use her faculties in the encounter with the self and the other. A person seeks connection with others in order to transcend her limitations, contemplating both the self and the infinite, and create something new. This desire to take part in Human Action and thus to emerge from the anonymity of the private is what it means to be free (Arendt 2018, pp. 186–87). Freedom implies self-consciousness and reflection, such that moral vision and creative impulse help the person in the shaping of her private environment and society (Bandura 2006). Exercises of freedom are in small and great ways acts of creativity—of creating various forms of property—and lie at the core of what it means to be human. The most meaningful aspect of temporal creativity is the intentional relationship of the person with other persons, which Arendt describes as occurring in the space between people (Arendt 2018, pp. 7–8, 182; 1968, p. 465).

Humans come to better know themselves in relationship with others and the physical environment. Personalist psychologist Carl Rogers defines the creative process as "the emergence in action of a novel relational product, growing out of the uniqueness of the individual on the one hand, and the materials, events, people, or circumstances of his life on the other" (Rogers 1995, p. 350). For Rogers, psychotherapy seeks to aid the natural process of growth and maturation by which the person simultaneously actualizes his potentialities while extending himself into his environment (Rogers 1995, p. 351). The creative process, as an essential human characteristic, means that a person, to be fully human, must develop a vision of herself and her society, and then intentionally work to actualize that vision.

The creative process is not a simple one-way dynamic of a single actor, either dominating others or compromising his own vision, diluting it with the vision of others, as Nietzsche or Sartre might claim. Like Rogers, Arendt points out that the creative process involves a tension between the creator as knowing himself as alone, separated from all others by virtue of this vision, but also urged to communicate this vision to others such that, without others, creativity and consequently personhood are crippled (see Arendt 2018, p. 76). Affect psychologist Donald Nathanson, following the work of Silvan Tomkins, describes the construction of the self, as formed by the self-consciousness of shame that both separates the individual from others but that then ties the individual to the one before whom he is shamed (Nathanson 1992, pp. 179–84).

The human person developmentally has no sense of the self without shame, which is the root of self-consciousness. For Arendt, the private sense of shame is necessary for the public sense of honor (Arendt 2018, p. 73). The sense of shame provides the person with an image of the self which captures deficiencies, and so motivates the self to improve for its own sake, for others, and for the public environment. The model for this is the family, which helps the person *individuate*—develop the distinctiveness of the self in relation to others—with the good of the family community as an end. The family is a model because of its normative role in the psychological developmental process; the family provides a sense of privacy which shields the individual as she matures and learns to interact with others. Naturally, this role should be adopted by the larger community in time, but what is important is that a space for privacy is retained, especially in psychologically developmental years. One critical experience that requires privacy is the experience of shame.

Shame usually carries negative connotations. Shame is inherently painful, but that does not make it bad. In the positive sense, it fuels the drive to create connections with others. This drive is crucial to human actualization and develops in conjunction with and for the benefit of society. The experience of shame is universal and necessary to form a sense of self and how to interact with others, beginning with the family.[11] Humans thrive when forming a mutual vision for society through open debate in the tension between shame and communion. Shame inculcates in the person an ordered sense of humility and

---

11 For a fuller explanation of the dynamics of human psychological development beyond the scope of this paper, see (Nathanson 1992), especially pp. 185–235; see also (Bandura 2006, pp. 164–80).

imperfection of the self and others. Shame is a fundamental affect, the building block of the complex emotion and disposition of humility. A sense of shame undergirds all personal relationships, in that persons constantly discern and navigate what is to be shared and what ought not be shared with another person. That sense is critical in compelling a person toward action.

Manu Samnotra in *Worldly Shame* describes Arendt's understanding of the role of shame in the mutual construction of a political world by an individual and the political community (Samnotra 2020). For Samnotra, it is the experience of powerlessness that generates shame initially and compels the individual to seek others who share a similar sense of shame in order to create a new order. Samnotra terms this occurrence "worldliness" (Samnotra 2020, chp. 1). Conversely, "worldlessness" occurs when the individual, in response to the experience of shame, diminishes the other, be it an individual, a group, or even a whole ethnicity, in order to maintain the stable social order and structure of political power.

Samnotra describes Arendt's biography of Rahel Varnhagen as an illustration of how a marginalized individual can participate in the worldlessness of the hegemonic class through doomed attempts at assimilation (Samnotra 2020, chp. 2). The marginalized person is required to accept an unequal status by continually distancing herself from her own identity in order to have some foothold in the hegemonic class. For Arendt, the proper social structure is one characterized by solidarity. Samnotra writes, "Solidarity, in the Arendtian sense, is the ability to connect and separate in a way that both tethers us to a community and maintains critical distance" (Samnotra 2020, chp. 3). Rahel had no solidarity with elite Prussian social circles because her maladjustment to shame, what Samnotra terms "shame-aversion", prevented her from recognizing any critical distance from that society, while at the same time, she was increasingly tied by that society to her poorer Jewish relatives, from whom she sought alienation.

The critical difference between Samnotra's understanding of the role of shame in building solidarity and that of Arendt centers on the dual role of shame and privacy. For Samnotra, sources of shame are only "ethical antennae" which alerts us to the need for others in order to build a new, more plural social order, and then, once that order is established, the shame lasts only as a residue that provides stability, guarding against disruptions of the new order it helped create. For Arendt, however, shame does not exist primarily as a public phenomenon, but as the part of the border between the public and the private. Shame has its source intrinsically in what should remain private. Shame is the mechanism that creates self-consciousness (Brendan Ignatius McGroarty 2006, pp. 62–63), and is coincidental to privacy. Neither privacy nor shame are incidental or occasional; both are intrinsic aspects of the subjectivity of the person, such that the compromise of either results in depersonalization. Samnotra's description of "desubjectification" and "resubjectification" is simply a description of shame operating on the borders of privacy, giving life to solidarity, not rigidifying social structures. It is not the shame of the tyrant that creates totalitarianism.

Nonetheless, shame becomes detrimental if the person denies its existence or withdraws from associating with others definitively, both of which isolate the person. For shame to develop appropriately into humility a sense of privacy must be maintained. Tyrants and factions of the citizenry can inflict a sense of shame on the person to make her conform; this is more effective if the citizen is deprived of privacy and humility is thereby inhibited.[12] The boundary between the self and other is guarded by shame, just as four walls protect the household and the border protects the *polis*. However, shame can be manipulated by the other to cripple the growth of the self.

For Arendt, the significant difference between the family and the public square is the shield of privacy in the former and the diversity of opinion in the latter (Arendt 2018,

---

12    This dynamic can be described variously as tyranny of the majority, mob rule, perceived tyranny of the majority, or groupthink.

pp. 39–40). Preparatory to public engagement, ideas may be inculcated and refined in civil associations, like schools or churches, and in the family. The human person comes to better know herself by interaction with others, and thus becomes self-conscious and free in a meaningful way. Humans associate with one another and negotiate the development of society as a part of normal psychological development. As we have seen, for Arendt, the private life is essential to maintaining the integrity of one's public life.

Associating with others helps a person shape her life, private and public (Neem 2006). In her private environment she pursues personal interests, including those pertaining to family, friends, education, spiritual practice, and so forth. Associating with others affirms her agency and status as a sovereign, decision-making person. It fosters productive humility by inculcating an appreciation of freedom and creativity in others. Through humility, the person sets aside private interests for the sake of the common concern, leading toward an Archimedean point from which the self can reflect on her life (Arendt 2018, pp. 257–68). Arendt illustrates this point metaphorically with the development of the satellite, which enabled humanity to look back to Earth and recognize the futility of clinging to its earthbound perspective (Arendt 2018, pp. 1–6). Humility gives us the freedom to know ourselves as others see us and to thereby interact with one another in a meaningful way (Arendt 2018, p. 268).

Associating with others helps achieve in the public realm what would otherwise be unattainable, thus leading to improved public safety, transportation infrastructure, schools, sanitation facilities, hospitals, etc. (de Tocqueville 2002, pp. 489–92 [vol. 2, part 2, chp. 5]). The desire to shape her life entails a need for an equal share in the political realm, and this is necessary for the ordered development of the self and of society. The equality of persons must be assumed on religious or natural law principles, but failing that it must be assumed. This can be assumed only where the right to privacy is guaranteed, because inherent in the right to privacy is the right to non-disclosure; the self is not exhausted in its expression through property, and since the self is not its own property, it can never be completely disclosed. This self is the source of creativity in property, and so it always holds a potential for participation in public action. Where that share is not equal or the dignity of privacy is not recognized, the person is subordinated to another person or class of persons, and thus, individuation is hindered, seeding alienation as a makeshift attempt to distinguish the self. In this way, the exercise of public freedom is a necessary activity of the human person.

## 4. Human Nature and Tocqueville's Types of Associations

The act of associating is part of human nature. As a result of this, associations pervade society.[13] Whether formal or informal, associations exist naturally as a product of the human desire to interact with others (de Tocqueville 2002, pp. 183–84, 496–97 [vol. 1, part 2, chp. 4; vol. 2, part 2, chp. 7]). People work through associations toward the common good and to sate certain individual needs, such as the acquisition of knowledge, socialization, spiritual development, and physical exercise. In pursuing these matters, people also hone their skills in the art of associating. Those skills enable one to accomplish things in conjunction with others: establishing schools, practicing religion, building parks, raising a family, and much more. They help one to function peacefully with others and exercise economic freedom, religious freedom, and a host of other purposes (Berger and Neuhaus 1977; de Tocqueville 2002, p. 492 [vol. 2, part 2, chp. 5]). To function in a civilized society, that is without violence, one must perfect the art of associating with others (de Tocqueville 2002, p. 492 [vol. 2, part 2, chp. 5]; Applebaum and Pomerantsev 2021). There can be overlap among associations in meeting various human needs, with more than one type of association meeting a specific need. For example, a need for friendship could be met through a gardening club, a church, or a sports club.

Many associations help build social trust amongst the citizenry by encouraging the participants to look outwardly and interact respectfully with people outside their usual

---

[13] (de Tocqueville 2002, p. 181 [vol. 1, part 2, chp. 4]). Tocqueville defined an association as the adherence by a group of individuals to a particular doctrine along with an agreement to cooperate with each other in a certain manner in order to make that doctrine prevail. See also (Arendt 2018, p. 22).

circle. This activity fosters an appreciation of the freedom and creativity in others and an understanding of their points of view, and it thus builds trust and wards off discord (de Tocqueville 2002, p. 492 [vol. 2, part 2, chp. 5]; Neem 2006, pp. 99–121). Such activity can inculcate a sense of shaping one's environment, though it cannot replace the benefit of participating in the permanent association. Within the sphere of an association, the person acts in concert with others, thereby furthering the association's mission. This builds the character of the person because she has advanced the mission of the association. The primary dynamic is the person's productive engagement with others, and its primary educative effects are produced by engagement with others, not didactic instruction from the institution of the association.

Tocqueville categorizes associations as the permanent, political, and the civil (Villa 2008, p. 34). Permanent associations are those recognized by law and designated as cities, townships, and counties (de Tocqueville 2002, pp. 57, 180 [vol. 1, part 1, chp. 5; vol. 1, part 2, chp. 4]). Political associations are those that seek to gain control over government and, to do so, engage in three types of activity: (1) gathering iterations of a particular opinion, refining them into clearer and more precise form, and expressing that opinion publicly; (2) assembling persons to engage in political advocacy; and (3) appointing certain individuals to represent them, as is done through a party's political conventions, caucuses, and primaries (de Tocqueville 2002, p. 181 [vol. 1, part 2, chp. 4]). The third category, the civil association, is a broad collection of the non-political, including commercial and industrial associations, as well as professional, garden clubs, sports, educative associations, religious associations, and a thousand other kinds of associations (de Tocqueville 2002, p. 489 [vol. 2, part 2, chp. 5]).

The permanent association uniquely meets certain human needs and provides certain benefits to society. Permanent associations are so called by Tocqueville because they form the framework for society itself and are universal in membership, even if active participation is limited. They are recognized by law and constitutions, and in that sense are created as legal entities. All other institutions and persons find their place within the framework of a permanent association. All associations bring their members together and imply a capacity to effect the end of that organization, but it is the permanent association in a democracy that implies this equality universally within its boundaries. To participate in a permanent association means to accept the equality of the other members of that association.

Political associations, which are distinct from permanent associations, provide a platform on which citizens can organize, refine the presentation of issues, propagate arguments, etc. They are also critical for the advancement of minority interests (de Tocqueville 2002, pp. 183, 248. [vol. 1, part 2, chp. 4; vol. 1, part 2, chp. 7]). Nonetheless, because the objective of a political party is to establish dominance over other parties as a means of correcting or forming public policy, they are poor vessels for the generation of broad social cohesion and are not effective at binding the people to the constitutional project. A political party is by definition partisan, not oriented toward inclusion or representation of the whole, even if their ultimate aim is good policy. In contrast, the association that most effectively generates broad, universal social cohesion is the permanent association and in particular the township or, in present parlance, local participatory government.

Tocqueville was fascinated by the natural rise of self-organized permanent associations in America. The early 17th century settlers in the American colonies were free to fashion democracy as they saw fit, unencumbered by its opponents "within the old societies of Europe" (de Tocqueville 2002, p. 12 [vol. 1, "Introduction"]), and unmolested by Britain during the era of its benign neglect. The colonists built structures of self-government through which citizens would have a share in the public business.[14] The most important of these was the township, which, if properly formed, is consonant with the human person's desire to shape her environment: "The township is the sole association that is so much in

---

[14] (de Tocqueville 2002, p. 40 [vol. 1, part 1, chp. 2]): The "township had been organized before the county, the county before the state, and the state before the Union".

nature that everywhere men are gathered, a township forms by itself". (de Tocqueville 2002, p. 57 [vol. 1, part 1, chp. 5]). The township forms the immediate physical environment of the person as he emerges from the shelter of privacy in the family and other associations. It should be noted that in Tocqueville's view, townships were simply geographic divisions of counties and states, reflected perhaps by local government, referring simply to local society itself. Despite the rise and perhaps distortions of social media and communications, the local township or area remains the first place the private person emerges into the public square.

Commentators frequently reference Tocqueville's examination of associations in discussing mediating institutions, but many of those examinations overlook the central role of the township in his analysis (Villa 2008, p. 30). Along with the family it is the institutional bedrock for building a sense of membership in the constitutional project and broad social trust amongst the citizenry. The township serves as a primary school for democracy and the exercise of freedom (de Tocqueville 2002, p. 57 [vol. 1, part 1, chp. 5]). Local decision-making brings citizens together and makes apparent the need for each other's support to get things done.[15] It is where people exercise their share of public power and see the results of their work daily. It is also where they do so in proximity to their family and friends. Beyond that, the township provides irreplaceable benefits to society and to the person because it is the key institution for binding the person to the constitutional structure and for generating solidarity among the citizenry.[16] Despite the rise of a mobile population and associations that are effected through modern communications rather than physical locality, local communities retain the character of permanence because it is only through them that personal presence can be conveyed.

The permanent association differs from other types of associations in its universality and in its relation to the constitutional structure. As to universality, the permanent association does not have geographic universality, but it does provide universality within its geographic boundaries. It also has a very low, arguably no, hurdle to membership. One need not apply for admittance, pay a membership fee, pass a professional test, demonstrate proficiency in a skill, participate in ritual or ceremony, or have a sponsor. By participating in it, the person becomes accustomed to interacting with all people on a take-them-as-you-find-them basis, rather than lapsing into a comfort zone of dealing with a narrow group. Through this, the person reaps the benefits of humility and shame, and strengthens the public forum. As Arendt observes, the first New England colonists, faced with entering a state of nature in unchartered wilderness, discovered they had the power "solely by the strength of mutual promise" to constitute all the laws necessary for government (Arendt 2006, pp. 158–59).

The keys are whether the township itself has independence and power and whether its citizens have an avenue for deliberation and decision-making (de Tocqueville 2002, pp. 63–65 [vol. 1, part 1, chp. 5]). If so, the person is naturally drawn to partake in the shaping of her environment through the exercise of her power. She seeks those outside her usual circle to understand their points of view, obtain their support in her view as to decision-making, and lend them cooperation (de Tocqueville 2002, pp. 485–88 [vol. 2, part 2, chp. 4]). Even where agreement is not reached, she has interacted with others in a peaceable way, often forming mutual respect and even friendships. This gives the citizen practical confidence in expressing her own opinions and in the *good* nature of her fellow citizen in respectfully considering those opinions. Whether others agree or disagree with a particular opinion, it is a forum for her to distinguish herself, and it further draws her outward to others. It is an antidote to the tyranny of the majority or perceived majority and

---

15 (de Tocqueville 2002, p. 63 [vol. 1, part 1, chp. 5]): "Now remove force and independence from the township, and you will always find only those under its administration and no citizens". and pp. 485–88 [vol. 2, part 2, chp. 4]: "Local freedoms, which make many citizens put value on the affection of their neighbors and those close to them, therefore constantly bring men closer to one another, despite the instincts that separate them, and force them to aid each another".

16 (de Tocqueville 2002, p. 301 [vol. 1, part 2, chp. 9]): Local government ensures that man's encounter with government does not "fall below the level of humanity".

to the tools of such tyranny, e.g., groupthink, corporatism, centralized government, and a centralized press.[17]

With the permanent association, the citizen sees every day the results of her participation and is inspired to do even more to participate in shaping her environment (de Tocqueville 2002, pp. 485–88 [vol. 2, part 2, chp. 4]). She sees political freedom as within her reach, becomes habituated to its use, and witnesses the tangible results (de Tocqueville 2002, p. 57 [vol. 1, part 1, chp. 5]; Arendt 2006, p. 110). That process results in multiple goods, the immediate ones of which are the decision made and the bridging between self-determination and a common spirit, which is necessary for a strong society. Furthermore, this activity invigorates the person's spirit, affirming her agency and encouraging her participation in civic, political, and religious organizations. Having a share of power to shape one's environment—public freedom—sates a human need, generating a feeling of happiness that could be acquired nowhere else (Arendt 2006, p. 110). That sense of agency in turn drives the person to seek out intermediary associations to engage in further Human Action. Conversely, centralization and the invasion of the private sphere quash the sense of agency, dispiriting the person from engaging in other associative activities and even enervating her private life. Public life consists of a constant balance of what must be centralized and retaining local or even private control when more central control unnecessarily distances individuals from the decision to be made.

Due to the significance of the township in encouraging participation in governance, it takes precedence, at least in historical terms, in the constitution of the nation. Hannah Arendt points out that the problem the French Revolution faced was the uncritical assumption that power and law came from the same source (Arendt 2006, pp. 156–57). Any Constitution written was composed by a source which itself was unconstitutional, and so could be superseded by the next decision of the legislature. In contrast, Madison proposed that the authority to compose the American Constitution derived from the states, whose authority was already locally established. In so doing, Madison set the precedence for subsidiarity, which holds that matters should be handled by the people themselves, or the closest authority that has categorical competence over the matter. For example, national defense is a matter for centralized government because it deals with external threats to the nation as a whole. Zoning issues are best handled by local communities as they are affected by the particular build-out of those communities.

The permanent association is also unique among associations with regard to the constitutional structure. Using the United States as an example, as constituent parts of a state's governance structure, the permanent association links to the national constitutional structure and is the government closest to the people. By participating in it, the person engages in the business of building society. This fosters calm, peaceable sentiments toward government, sentiments of membership *in* government rather than being the subject *of* government, while, at the same time, developing solidarity among the people as noted above. As Tocqueville observed, from a political standpoint, America's decentralization ensures that the citizen applies himself to each of the country's interests as if they were his own. He is "glorified in the glory of his nation; in the success…he recognizes his own work and he is uplifted by it".[18]

In "All for All: Equality, Corruption, and Social Trust", Bo Rothstein and Eric Uslaner dispute the popular notion that a lack of participation in civic organizations underlies declining levels of generalized trust (Rothstein and Uslaner 2005, p. 53). We agree, but for different reasons. As noted above, general civic engagement, that is engagement other than in the permanent association, provides practice in organizational and socialization skills. However, it does not bind people to a national project. In contrast, having a share in

---

17   See (de Tocqueville 2002, pp. 407–10 [vol. 1, part 2, chp. 2]): discussing the dangers of public opinion in a democracy.

18   (de Tocqueville 2002, pp. 82–93; 220–27 [vol. 1, part 1, chp. 5; vol. 1, part 2, chp. 6]). Quotes in this paragraph are from page 89.

the exercise of power, that is, being part of a body that exercises power, binds one to the constitutional structure and universally binds one to other citizens.

In the absence of local decision-making, one might classify another person as not like oneself. However, if one is working with others to exercise power, those others subjectively become like oneself: a citizen engaged in a common project. In exercising that power, the person is constantly reminded of the need for others theretofore not like oneself.[19] This builds broad trust in society and invigorates the person's confidence that she has agency over her life. In brief, we agree with Rosthstein and Uslaner that equality is the key to building trust, but are of the view that, next to equality under the law, political equality—having power and an equal share of it—is the key form of equality.[20] When all people have an equal right to exercise their political freedom, then the practice of politics demands that all persons be acknowledged as equal in their essential characteristics. That dynamic pulls the fallible person away from re-classifying accidental differences as substantial. In this way, political equality builds trust.

## 5. Selfishness, Individualism, and Freedom

Free societies are often described as selfish and materialistic, especially when a free economy is the focus of discussion. That charge rests on a misunderstanding of the dynamics of public freedom. To begin, distinctions must be drawn among selfishness, individualism, and individuation. *Selfishness* means lacking concern for others and is chiefly motivated by a desire for personal pleasure and gain, although, as noted below, other habits can encourage that desire. *Individualism* or *individualistic* has several meanings in common usage, some with positive and some with negative connotations. For the purposes of this work, individualism refers to the tendency to turn one's associative activities inward to the care of family and friends. A tyrant strives to create individualism, if not selfishness, to isolate voices that may challenge his own, and turn them away from the public sphere (de Tocqueville 2002, p. 485 [vol. 2, part 2, chp. 4]). One who suffers from individualism has withdrawn from concerning herself with the good of society and those outside her circle of family and friends. She may participate in society to a degree, and even do so enthusiastically, but even then, she hides her true self, afraid to express her most profound sentiments because she does not trust her fellow citizens and thus fears the tyranny of the majority.

Individualism is not selfishness. Rather, it is often a reasonable, but not ideal, reaction to the choices presented to the person. It is:

> a reflective and peaceable sentiment that disposes each citizen to isolate himself from the mass of those like him and to withdraw to one side with his family and his friends, so that after having thus created a little society for his own use, he willingly abandons society at large to itself. (de Tocqueville 2002, p. 482 [vol. 2, part 2, chp. 2])

Individualism can arise in a politically battered person who retreats in frustration to the smaller community of self, family, and friends. It can arise where tyranny of the majority reigns, suppressing free discourse. In modern parlance, it can arise where groupthink reigns, suppressing the person's natural actions to individuate herself. It can arise as well in a person of unfortunate circumstances who is struggling day-by-day for any of a variety of reasons. It can also arise where centralization of one type or another has ravaged political freedom and foreclosed authentic participation in governance. Individualism is to be avoided because it steers away from bridging opportunities, because its practice leads

---

[19] (de Tocqueville 2002, pp. 57, 486–87 [vol. 1, part 1, chp. 5; vol. 2, part 2, chp. 4]). The Framers found it "fitting to give political life to each portion of the territory in order to multiply infinitely the occasions for citizens to act together and to make them feel every day that they depend on one another" and "Thus by charging citizens with the administration of small affairs, much more than by leaving the government of great ones to them, one interests them in the public good and makes them see the need they constantly have for one another in order to produce it".

[20] (de Tocqueville 2002, p. 479 [vol. 2, part 2, chp. 1]): "Equality can be established in in civil society and not reign in the political world".

toward selfishness, and because it is contrary to human nature (de Tocqueville 2002, p. 483 [vol. 2, part 2, chp. 2]). Individualism in the sense we are using the term reigns in tyrannical states such as the former Soviet Union, which strive to invade the person's privacy, crush her soul, narrow her circle of trust, and limit her individuation. Individualism is characterized by the condition of isolation, which for Arendt is the impotence of persons engaging in Human Action, an occurrence of which can result in conformism (Arendt 1968, p. 474; 2018, pp. 38–49).

Unlike individualism, individuation, or personalization, is good. A person is an individual with self-consciousness, which enables her to come to know herself by engaging in self-reflection. That process begins with interactions of appropriate shame or esteem with one's family and close associates. As the person grows from self-consciousness to a larger consciousness of society, participation in the larger sphere becomes meaningful and necessary to maintain this growth. Nevertheless, the person actuates herself through her local environment, most acutely with those whom she knows personally. Only derivatively does she extend this interaction and freedom to the larger society of individuals whom she has never met.

A person knows herself as unique, or individuated, not simply to stand out against the background of others, but as a sense of the good that she might uniquely produce for those in her immediate environment and toward the improvement of that environment. In the most profound sense, this describes a person of property. In providing these goods for others, she necessarily becomes intimately aware of the uniqueness and potential of others. Arendt holds that the original sense of property was that which was attached to the soil, that is, immovable and non-commodifiable, and therefore apart from the public realm and could serve as a sanctuary from it (Arendt 2018, pp. 61–62, esp. n.56). As she is unique, a person is not replaceable. If she is taken away from her community, or if she is segregated within it, that community is altered and its integrity diminished, no matter what might be gained by the addition of others. Collectivists, elitists, and totalitarians seek to invade the person's privacy, to isolate her, and to force her conformity.

The tendency of government is to increasingly grab power and centralize it. As state and federal government supersede more local government, the person is relieved of having a meaningful share in shaping her environment. Decisions are now too remote for her to have any effect. The person then lacks a compelling incentive to work with others and to understand them. She nonetheless continues to affirm their equality because it affirms her own equality. However, without the practice of understanding the subjectivity of others and forming bonds with them, she loses confidence in the good nature of the citizenry. She then shies from individuating herself—making known her true views and pursuing happiness–and narrows the exercise of her reason. The sense of shame, which in a healthier context compels her to improve herself and society, instead cripples her inclination to make her ideas known, since she cannot improve society in so doing. Shame then becomes an interior sickness, isolating her and deadening Human Action. She strains to affirm the groupthink of perceived majority opinion. In this way, centralization perverts democracy and induces the person to surrender her individuation (de Tocqueville 2002, p. 410 [vol. 2, part 1, chp. 2]; Arendt 2018, p. 58; 1968, pp. 465–68).

Regardless of having a share of power, should the person nonetheless reach out to others unknown to her, to enlighten them, to be enlightened by them, and to build trust? In the abstract, the answer is certainly in the affirmative. However, such a judgment can be uncharitable because the return-on-time-and-effort analysis differs for each person. Harkening to Arendt's tripartite classification of the active life, for some people time pressures might always require an inward focus, such as those who work multiple jobs to make ends meet and those who have time-sapping family duties such as caring for a sick child. Arendt describes the life of the laborer, as opposed to the worker, as one enslaved to necessity, unable to produce any lasting good that survives the metabolism of life (Arendt 2018, pp. 81–84). Similarly, the parent who contends with an unsafe neighborhood may need to focus inward to rely on her circle of friends and family to forge a safe environment

for her children. Other scenarios abound under which a person may understandably need to focus inward.

Note that, in this last hypothetical, the parent stands in a predicament. A genuine outward effort of common cause might indeed generate a strong associative effort that solves a problem, but can anyone rationally second-guess the strategy and intentions of the mother who elects to look inward to family and close friends to solve the day-to-day struggle on behalf of her child? Is this not especially so if authority has disempowered her so that she reasonably doubts the efficacy of outreach to strangers? The predicament is not one of her own making.

If equality of conditions reigns, then the individual will necessarily have an equal share of public freedom—an equal share of power in participating with other citizens in the shaping of her local environment; in other words, an equal share in pursuing public happiness. However, equality of conditions can be dangerous if combined with a sense of powerlessness to affect the environment outside her immediate surroundings. Equality can imbue the person with a sense of complacency—that she owes nothing to others, that she expects nothing from anyone, and that other citizens are her equal and therefore not a threat. It encourages her to embrace the idea that she must act and think of herself as independent from others. Thus, she falsely and complacently perceives a diminished need to know others (de Tocqueville 2002, pp. 483–84 [vol. 2, part 2, chp. 2]). She might interact with them superficially, but she is not inclined to understand their subjectivity nor to encourage them to reciprocate (de Tocqueville 2002, p. 410 [vol. 2, part 1, chp. 2]). That dynamic isolates her from others, inhibits the formation of bonds with them, and thus leaves her with a sense of powerlessness. Centralized government prefers citizens to have this sense of complacency so that the citizenry does not contest its pronouncements.

The right to shape one's private life and the right to have a share in shaping the public domain draw from the same personal reserve of time and effort. In deciding how to parse her activities, an individual will naturally engage in a return-on-time-and-effort analysis. Activities likely to bear fruit will garner a higher share of one's time and effort, and those unlikely to bear fruit will be curtailed (Bandura 2006, pp. 164–80, 170). That common reserve creates a fragility. On one hand, if politics or some other force removes opportunities for the individual to participate in shaping her environment, she will turn inward to her private life to address those things that she can influence. To the ill-informed observer, she may appear to have become selfish, but in truth, an antisocial rupture has occurred that has radically diminished her efficacy in the public square and made it sensible that she instead focus on doing good in a smaller, more private circle.

In a decentralized society, one with public freedom, the person can follow her natural yearning. She is drawn outward to know others, to further shape her environment, and to draw her friends and family, particularly her children, outward toward the beauty of participating in a wider circle. Observed from the outside, the individual will seem to be spurning the inward turn of individualism, but from her perspective, the permanent association has extended an invitation to exercise her share of power. She is actuating herself through the use of her property by enhancing the mutual environment of others, advancing in self-consciousness, and encouraging her circle of friends and family to do the same. As we have seen, property consists of those intellectual, physical, spiritual, or other elements of her personal environment which she has shaped and inscribed privately and which she might display or use in the shaping of her public environment.

In viewing the United States in the context of this issue, Albert Bandura has suggested that Americans are simplistically categorized as individualistic (Bandura 2006, pp. 174–75). Such judgments imply that Americans are overly concerned for their personal welfare relative to the greater good. Bandura shows that people of any culture participate in outwardly focused projects and discussions about the common good according to the degree to which they feel efficacious in so doing. Building on Bandura's analysis, American individualism is largely a function of the lowered efficacy of local participation.

## 6. Participation, Third-Party Esteem, and Constitutional Confidence

If the weapons of totalitarianism are disordered shame and psychological isolation, then the instruments of a constitution must encourage participation from the complete range of society, not privileging elites or technocrats nor diminishing local concerns. The citizens must be drawn out from their immediate concerns to appreciate larger concerns that touch upon what it means to be human, and the moral direction of their society. Whatever the value of expertise, moral judgment is common to all, and the development of character and virtue is a project for all.

Writing a quarter century ago, Jane Mansbridge traced the line of thought concerning political participation beginning with a concern to develop character because that would lead to a healthy *polis*. Aristotle contended that the *polis* had "to devote itself to the end of encouraging goodness" and discussed sound laws and a good polity as furthering the development of personal character, justice and goodness, which would in turn contribute to better decision-making. Machiavelli emphasized internal conflict, external debate, and refinement of the law as enabling the individual to better contribute to the polity and in that process derive certain benefits, such as the acquisition of knowledge, all for the good of the *polis*. Rousseau espoused good laws and a civil religion, to substitute a sense of justice for man's raw instinct and a sense of duty for his appetite; this would help the individual recognize the common good and act upon it. The person's character, in turn, would be improved by willing good law (Mansbridge 1995, pp. 1, 4–7).

Mansbridge notes, "It was Alexis de Tocqueville, returning from America, who first claimed that participation in the process of governing developed individual character". (Mansbridge 1995, p. 4). Tocqueville's claim arose most directly from his observations of the New England town meeting, a dynamic that Mansbridge describes as "ownership through power". Mansbridge notes that John Stuart Mill, building on Tocqueville, recognized that participatory democracy produces goods in addition to the policy decision made and that these goods accrue to the participator and derivatively to society. In our analysis, we identify the goods produced in the person by participatory democracy as: the good arising from the decision made; the good to the citizen from having a share of power in the decision-making; the good arising from the relationships formed or made by those involved in the decision-making, which in Arendt's framework is created in the space between individuals. Those goods derivatively generate third-party esteem, increased trust in the citizenry and in government; and a binding to the constitutional structure (Mansbridge 1995, pp. 1, 4–7). As Arendt noted, participation in governance is an activity that people enjoy teleologically; that is, it touches upon the most innate desires of human nature (Arendt 2006, p. 110). In addition to having the merit of furthering one's own interests, it brings happiness, or the fulfillment of human *telos* as a social animal.

As a complement to the desire to socialize, the human person has "a desire to be observed, considered, esteemed, praised, beloved, and admired by his fellows". (Adams 1805, no. 4; Arendt 2018, pp. 196–99).

> Wherever men, women or children are to be found, whether they be old or young, rich or poor, high or low, wise or foolish, ignorant or learned, every individual is seen to be strongly actuated by a desire to be seen, heard, talked of, approved and respected by the people about him, and within his knowledge. (Adams 1805, no. 4)

One tends to respect the rights and dignity of others not just due to notions of reason and justice but also because one wants to be esteemed for having done so. It is a reward furnished by nature for purposes of "promoting the common good, as well as respecting the rights of mankind" (Adams 1805, no. 4). This desire for esteem serves to reinforce the need to recognize the rights and dignity of others.

An individual who is engaged in an association's conversations, deliberations, and decision-making tends to enjoy the esteem of fellow members. Members confer esteem on each other, and such showering of esteem reflects their own status. A well-known example of this dynamic is United States senators' tradition of conferring esteem on fellow

members, even those from other parties. Participating in the process and abiding by the rules and conventions of the association garners prestige for the individual members. Such conferral of esteem has a far more significant role in society than just the stroking of egos. It is a way of aiding the process of self-reflection and self-consciousness, enabling one to recognize the importance of her activity in the well-being of others, in the same way that the withholding of such esteem might create shame and a corrective to self-perception and purpose (Nathanson 1992, pp. 250–52). As with associating generally, the benefits tend to become more robust with an increase in their exercise. Just as social and monetary benefits are enriched with increased economic association, so too do social and political benefits tend to increase with participation in local government.[21]

We use the term *third-party esteem* for the dynamic in which the esteem showered on a member becomes known by persons whom we call *third-party beneficiaries*: that person's relatives, friends, and neighbors who identify with that person. A similar notion, *recognition*, has merit but differs significantly from the term third-party esteem as we use it (Iser 2019). In that usage, "recognition" refers to the positive reception an individual or group has in order to participate in society. Similarly, in that usage, the term "esteem" refers to the intentionally positive recognition given to a group that may have been marginalized in the past. The difference between this usage and our usage of the terms related to "third-party esteem" is that the latter terminology refers specifically to the process by which an individual emerges from privacy into the public sphere by virtue of his or her relations with someone already in the public sphere. It is fully intentional, and can be done to varying extents, depending on how much of the person or property is to be publicly disclosed. The emerging citizen may vote anonymously, or submit a letter to an editor, yet choose not to be recognized. This is because one's public identity, and all information about oneself, is part of property, and must be protected by the shield of privacy; recognition is not necessary if that is not willed. The shield of privacy guarantees equality.

One tends to develop trust in an association that esteems one's relative or friend. Associations confer esteem on their members as a by-product of their participation. When an association does this, the member's family and friends develop an affinity for the association through third-party esteem. The converse happens when the association confers contempt on a member. *Third-party contempt* occurs when a community holds a member in low esteem by, for example, ignoring her or excluding her from decision-making and that fact becomes perceived by someone who has an affinity for her through the weaponization of shame. Third-party contempt explains much of the disaffection for government and some of its pillar institutions. In these ways, the positive and negative effects of the association extend beyond its members.

Several factors contribute to the social trust that arises from third-party esteem. It naturally increases as the strength of the bond between the member and the beneficiary of third-party esteem increases, a dynamic that myriad policymakers overlook when lamenting matters such as the decline of the family or juvenile crime. Likewise, it increases with the greater importance of the underlying activity in the eyes of the beholder. A consistent conferral of esteem leads to stronger third-party trust than a fleeting incident.

In the context of the permanent association, several points are noteworthy. One, a consistent conferral of esteem leads to stronger third-party trust than a fleeting incident. Two, third-party esteem binds the third-party beneficiary to the community and local government. Three, through his affinity with an association's member, the third-party beneficiary develops an affinity with the permanent association and, with that, the larger constitutional structure in the same way that the direct beneficiary binds to that structure.

---

[21] (de Tocqueville 2002, pp. 57; 486–87 [vol. 1, part 1, chp. 5; vol. 2, part 2, chp. 4]): The founders found it "fitting to give political life to each portion of the territory in order to multiply infinitely the occasions for citizens to act together and to make them feel every day that they depend on one another. . .. Thus by charging citizens with the administration of small affairs, much more than by leaving the government of great ones to them, one interests them in the public good and makes them see the need they constantly have for one another in order to produce it".

The permanent association's qualities of universal membership will tend to broaden the trust developed in the third-party; as the third-party becomes aware of diverse members showering esteem on his family member or friend, he will naturally be more open to the wider community.

Third-party esteem's function of broadening trust becomes particularly important for intergenerational trust-building and for the creation of strong sentiments toward the constitutional structure. The parent–child bond builds in the child a desire for the parent to be esteemed by others. The dynamics of the permanent association can prepare the child to bridge society and help cleave him to the overall constitutional structure.

From the standpoint of society at large, these dynamics help account for the declining cohesiveness of society. From the standpoint of individuals, these dynamics are likely most consequentially experienced by marginalized persons because those of greater material means can tap into fabricated esteem, albeit not universal in character. For instance, they can join the closed environment of a country club.

If it is stripped of independence and power, the permanent association becomes a mere administrative tool of centralized government. Consequently, the individual has a lower return on time and effort and thus less reason to participate. The incentive to reach out to others dissipates, and rather than fostering a feeling of membership in the overall constitutional structure, the permanent association is likely to generate a feeling of distance and alienation from it. Centralization of decision-making strips the citizen of any sense of ownership and makes her indifferent to the place she inhabits. Given the personal growth flowing from having a share in shaping society, it should be expected that depriving the citizen of that opportunity is dehumanizing.

Arendt observes that, since the revolutions of the 19th century, powerful government administrators are selected in the same way that scientists are hired; that is, not by widespread acknowledgement but by recognition from other elites (Arendt 2006, pp. 269–71). The practice of governing becomes no longer the result of participation but the nonpolitical decrees of experts. The common objection to the reinvigoration of participatory democracy is that the modern world cannot accommodate such decision-making. The world is much more complex than in the 1800s; in the face of such complexity, government needs experts, and the population has increased exponentially, which, as the narrative goes, makes participatory democracy impractical. Such arguments do not account for advances in technology and the information age. For example, modern communication technologies have not been applied to the practice of participatory democracy. To this end, various initiatives are under way to reinvigorate the practice of democracy and to soothe discord and alienation (Applebaum and Pomerantsev 2021). The problem is one of disuse and unfamiliarity, not one of impracticality.

Excluding people from having a share in shaping their local environment "is fit only to enervate the peoples who submit to it, because it constantly tends to diminish the spirit of the city in them" (de Tocqueville 2002, p. 83 [vol. 1, part 1, chp. 5]). Most especially, being pushed away from a natural activity demoralizes the person and denigrates her in the eyes of those who have affinity for her. If government or another force usurps a parent's share in shaping her environment, then it has frustrated the parent's efforts to lead her family, narrowed her efficacy, and it has done so in the eyes of her children and crippled the constitutional project.

## 7. Government and the Defense of Human Dignity

Jane Mansbridge noted that social scientists cannot prove that participation makes for better citizens. Those who participate feel it, and those who observe that participation believe it, but the "blunt instruments of social science" cannot measure the subtle changes in character that come about slowly (Mansbridge 1995, p. 1). Nor, it might be added, can they evaluate the character of a child raised in an environment of ordered democratic participation. Unfortunately, that intangibility makes it easy for policymakers to centralize decision-making.

Looking again at the United States, one sees a mere nod to the participation of the individual. State and federal central planners craft cooperative agreements and grant offers (collectively grants or grant offers) to the states or localities and attach procedural conditions under which the state or locality is to accept the programs. Those conditions regularly dictate which state or local official will make or reject the offer and the internal consultative process that official must employ. Central planners will also deploy conditions that shape the external consultative process, dictating the groups with which a state or local official must meet and imposing requirements for public hearings and notices. They dictate that policies be changed, statutes enacted, that certain authorities be vested in certain officials and that other officials provide an official opinion on the grant offer.[22] They require the formation of commissions and boards and require that those entities be invested with certain duties. They require states to form comprehensive plans (state plans) in broad subject areas, as has been done in education and transportation. These requirements are inserted in grant offers to create a favorable pathway for acceptance of the grant and the attendant substantive policy strings and in order to ensure the grant and its attendant strings have statewide effect. However, such process centralization (1) changes the state or locality's deliberation and decision-making processes, often compromising checks and balances, (2) can dictate a usurpation of local decision-making in favor of centralized state or federal decision-making, and (3) further distances the citizen from decision-making, thus inducing individualism and other pathologies.[23]

Through these process conditions, centralized government imposes policies onto communities by fracturing state or local government's consent system or by circumventing the habits and conventions of a community.[24] In the effort to regulate local communities and ostensibly eliminate corruption, local decision-making is paternalistically eliminated and new avenues of corruption and inefficiencies are created, all because local officials and citizens are not trusted with their own governance. Centralized government exerts its influence through power, money, or both, and it does so to affect education, transportation, policing, transportation, zoning matters, and much more. Andy Smarick summed up the effects of process intrusion into a community in a policy paper on philanthropy, education, and urban communities. His thesis holds for interventions by centralized government as well as for those by philanthropy: such interventions can cause "massive disruptions and reordering of the community's institutions", disempower citizens, and destabilize the community (Smarick 2021).

In a 1996 *Philanthropy* essay, Richard Cornuelle described this mindset from the citizen's perspective. It rests on the belief that management of a resource or service can only be conducted through monopolization of the resource or service. This belief implies that such management requires an external or centralized authority and that it would have to be carried out by specialized professionals. Cornuelle decried this "irrational disconnection of ordinary people from the business of the society, a radical constriction of the definition of the citizen's role". This, he argued, stood "the secular version of the familiar doctrine of subsidiarity" on its head. That perversion "is the real root cause of the evident loss of the feeling of cohesion and solidarity" (Cornuelle 1996).

If we accept the premise that the constitutional structure rests on the idea that government's authority flows from the people, who delegate a portion of their sovereignty to federal and state government, then government should indeed tread carefully and not step on the person's reservoir of sovereignty and power.[25] Apart from the duty it owes to

---

22  Bridget A. Fahey, "Consent Procedures and American Federalism", 128 Harvard L. Rev. 1561 (April 2015). In this article, Fahey coined the term "consent procedures". The article explores the federal government's attaches conditions to its grant offers to the states that dictate what state official or body will accept or reject the offer and the process that official or body must use in making that decision.

23  See (Hamburger 2021; McGroarty et al. 2017); Fahey, "Consent".

24  See Footer 23 above.

25  James Wilson, Pennsylvania Ratifying Convention (4 December 1787), *The Founders' Constitution* 1, chp. 2, doc. 14 (The University of Chicago Press), https://press-pubs.uchicago.edu/founders/documents/v1ch2s14.html (accessed on 14 August 2023): Sovereignty "*resides* in the PEOPLE, as the fountain of government; that the

a state, the federal government owes a special duty to a state's *citizens* to ensure that its actions do not truncate the state's decision-making processes. As the Supreme Court stated in a unanimous decision, federalism requires more than just setting boundaries between state and federal government to uphold the integrity of those institutions. It also directly secures certain liberties for individuals.[26] Some of these liberties are political in character, such as ensuring that a citizen has agency in shaping her environment. Given that those individuals are also federal citizens, the federal government should refrain from attaching burdensome *procedural* conditions to its grant offers to state and local government.

Such conditions frequently wreck the state's constitutional decision-making process, diminish the individual's political efficacy in state and local government, and further alienate the citizen from government. On the front end, through the grant's substantive policy strings, centralized government can pick policy winners and losers. On the back end, the grants displace political accountability, making state and local government bear the brunt for bad policies.[27] Such disempowerment spurs the human person toward individualism, and generates cynicism toward the constitutional structure and government at large (Hamburger 2021, pp. 108–10). Moreover, contrary to Chief Justice Roberts's admonishment, the state, with its decision-making process compromised, is often unable to simply refuse the "federal blandishments when they do not want to embrace the federal policies as their own".[28]

## 8. Conclusions

Hannah Arendt's constitutionalism calls for consideration of how the state encounters the human person. The human person is free and capable of meaningful association with others. Through property, the person expresses herself and interacts with others, and thus ownership of property is a fundamental, very personal, human right. Inherent in that ownership is a private realm wherein the person can reflect, build herself, and have a "location" in the world. Far from causing the person to turn inward to individualism or selfishness, property and the privacy that arises from it give the person a platform for even more powerful creation: the association with others for greater achievements, those which cannot be accomplished, in Arendt's terminology, through mere labor or work. This possibility sates the human yearning to shape her environment, encouraging the person to look outward to understand others not like herself and to seek their assistance in building society and to inculcate her family and friends in that activity. That exercise bends society toward tranquility. Third-party esteem, a good that flows from Arendt's constitutionalism, gives a person's family, as well as her friends, witness of her being held in high regard by constitutional society. It leads to trust in the constitutional system, tranquility among the citizenry, and strengthened bonding within the family.

The crisis of modernity is the suppression of Human Action, denying the individual an equal share in governance and invading her private sphere to isolate her and make her uncomfortable even in her contemplative life. Authority needs to be centralized to deal with the complexities of modernity, or so the tyrant and the collectivist contend. This deprives the person of power, making it seem that more good can be done by focusing inward to care

---

people have not–that the people mean not–and that the people ought not, to part with it to any government whatsoever. In their hands it remains secure. They can delegate it in such proportions, to such bodies, on such terms, and under such limitations, as they think proper".; Alexander Hamilton, *The Federalist Papers*, no. 22 (14 December 1787), reprinted in *The Founders' Constitution* 1, chp. 5, doc. 23 (The University of Chicago Press), http://press-pubs.uchicago.edu/founders/documents/v1ch5s23.html (accessed on 14 August 2023) ("The fabric of the American Empire ought to rest on the solid basis of THE CONSENT OF THE PEOPLE. The streams of national power ought to flow immediately from that pure original fountain of all legitimate authority".); James Madison, The Federalist, no. 37 (11 January 1788), reprinted in *The Founders' Constitution* 1, chp. 9, doc. 9 (The University of Chicago Press), http://press-pubs.uchicago.edu/founders/documents/v1ch9s9.html (accessed on 14 August 2023).

26  *Bond v. United States*, 564 U.S. 211, 221 (2011) (citations omitted).

27  See Footer 23 above.

28  *National Federation of Independent Business v. Sebelius*, 567 U.S. 519, 579 (2012) (Roberts, op.), *citing Massachusetts v. Mellon*, 262 U. S. 447, 482 (1923).

for family and friends. Her neighbors become strange, depriving her family and friends of third-party esteem and inviting a lack of trust in the constitutional system. Disorder does not stop there. Tyranny invites totalitarianism, bit-by-bit, and it seeks to invade the privacy of the person, depriving her of an earthly location and trying to substitute its views for hers. Once degraded, the person lapses in the protection of her privacy, making the private realm ever-more susceptible to incursions from government as well as others.

**Author Contributions:** Conceptualization, B.M. and E.M.; methodology, B.M. and E.M.; validation, B.M. and E.M.; formal analysis, B.M. and E.M.; investigation, B.M. and E.M.; writing—draft, review and editing, B.M. and E.M.; project administration, B.M. and E.M.; funding acquisition, E.M. All authors have read and agreed to the published version of the manuscript.

**Funding:** This research was funded by grants from The Lynde and Harry Bradley Foundation, grant numbers 20221534 and 20211384. There was no APC for this article.

**Conflicts of Interest:** The authors declare no conflict of interest.

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
