# Peer review of "Privacy, Property, and Third-Party Esteem in Arendt’s Constitutionalism"

_laws, 1980_

Round 1

Reviewer 1 Report

This essay offers a promising contribution to debates on Hannah Arendt's constitutional thought through the emphasis on the interplay between the private and the public and the role of property therein. It is relevant to the contemporary constitutional discourse particularly through (1) its engagement with the role of public participation for the sustainability of constitutionalism (although this comes largely only at the end of the essay) and (2) its thinking on the role of property rights. 

Although the essay is generally readable and incorporates several relevant references, there is room for improvement, with my central recommendation being the clarification of the connection between property and the private sphere. These two domains cannot be automatically equated, with the private sphere attributed primacy in Arendt. Minimum standards of sufficiency, albeit they well may have to be met for there to be a constitutioanl system, do not amount to the primacy of the private over the public, especially since it is through the public sphere that their guarantees operate. In this context, the references to the dominance of the family are less convincing as well (e.g. l. 303-304). Because of that, I would recommend to disentangle more clearly the relationship, including potential contrasts, between (types of) property guarantees, and the private/public divide. I would also recommend to consider using 'recognition' instead of 'esteem', the former being a widely discussed concept in political philosophy that appears to overlap with the authors' concept of esteem (or at least clarify the distinction between the two in the authors' understanding); to reduce the number of sections which currently makes the essay read at times as a more fragmented one; and to bring in the important and very interesting thoughts on Arendt and participatory democracy (l. 657 et seq., l. 713 et seq.) earlier on into the discussion. 

I have attached more specific comments to this review (including with some highlights which serve to indicate some of the key or most interesting points in my reading, for reference). I invite the authors to consult these and wish them success to develop the manuscript which I think has promise to become a useful contribution to (republican) constitutional theory. 

The English is generally fine, though there are some typos that would need rectifying. Furthermore, the structure (too many short sections) hinders the comprehensiveness of the text, so I suggest to reduce the number of sections and improve the transitions between them. 

Author Response

We would like to thank you for your generous response to our essay on Arendt's Constitutionalism. We have gone through the text and made changes in response to all the particular areas over which you expressed concern, and have significantly restructured the paper, reducing the number of sections and providing the reader with a sketch of our line of thinking in the introduction.  Please find attached a revision of the paper. It is now about three pages longer, and nine sections shorther. We believe significantly better thanks to your insight.  Any sections that were added are in yellow highlighter, as are sections that have been moved.  Little has been deleted except the removal of redundant points and excess verbage. Below is a response to each of the points in your review. Again, thank you for your help.

From the body of the review:

Although the essay is generally readable and incorporates several relevant references, there is room for improvement, with my central recommendation being the clarification of the connection between property and the private sphere. These two domains cannot be automatically equated, with the private sphere attributed primacy in Arendt. Minimum standards of sufficiency, albeit they well may have to be met for there to be a constitutional system, do not amount to the primacy of the private over the public, especially since it is through the public sphere that their guarantees operate.

We agree with this observation, particularly from the perspective of Arendt. We did not mean to imply the private has primacy over the public, just perhaps that the private precedes the public from a logical or perhaps developmental standpoint. The public sphere is constructed, and the private is natural, or perhaps pre-intentional.  We have consolidated some of the references to property, and describe it as the means by which the person emerges from the private sphere into the public.  But as you point out, the public protects the private, as the private protects the public and shelters both the self and property.

In this context, the references to the dominance of the family are less convincing as well (e.g. l. 303-304).

References to the family serve as a normative description of the context of privacy.  Associations tend to be first and closest with family members before being extended to more public figures. It also tends to be the context in which ideas are tested before entering public fora like schools, clubs, churches, and perhaps politics.  No normative description is implied; the family simply means those closest to a person who garners trust.

Because of that, I would recommend to disentangle more clearly the relationship, including potential contrasts, between (types of) property guarantees, and the private/public divide.

As stated above, in the revision we have attempted to clarify the relationship between property as a mediator of the self in the private/public divide.

I would also recommend to consider using 'recognition' instead of 'esteem', the former being a widely discussed concept in political philosophy that appears to overlap with the authors' concept of esteem (or at least clarify the distinction between the two in the authors' understanding);

This is a particularly fascinating observation, which will require more intensive reflection and observation by the authors.  We have added a tentative explanation of the distinction between third-party esteem and recognition.  Essentially, we believe the additional terminology is still warranted because recognition is a public term, and seems to entail the disclosure of some aspect of the person being recognized.  Third-party esteem is not a term that requires public disclosure, and typically has its origins in the private sphere, and is not referring to a relationship that can be expected automatically in a public relationship, as recognition is.

to reduce the number of sections which currently makes the essay read at times as a more fragmented one;

the number has been reduced by half

and to bring in the important and very interesting thoughts on Arendt and participatory democracy (l. 657 et seq., l. 713 et seq.) earlier on into the discussion. 

This has been added to the introduction and included at various other relevant points throughout the essay.

Reviewer 2 Report

It is rare to come across new contributions to the literature of justificatory theories for property.  This article makes such a contribution.  It is, simply, an outstanding piece of scholarship.  What more needs be said?  It should be published with the highest priority and the author(s) congratulated for their work.  If I was to make one suggestion, it would be this: there are a number of sub-sections to the article; it would be helpful to add a paragraph or two to the end of the current introduction that outlines what the paper will do--providing a roadmap to the sub-sections.  But this is not a necessary change.

Author Response

We thank you for your generous review of our essay.

In response to your suggestion, we have reduced the number of sections from 18 to 8 by combining many of the sections, though we did add about three pages in explaining some concepts we neglected.  Very little was deleted.  Everything that was moved or added is highlighted in yellow.

We also added two paragraphs as a synopsis of each of the sections to the end of the introduction.

Thank you for your help with the paper.